# Effect of Pectin/Nanochitosan-Based Coatings and Storage Temperature on Shelf-Life Extension of “Elephant” Mango (*Mangifera indica* L.) Fruit

**DOI:** 10.3390/polym13193430

**Published:** 2021-10-06

**Authors:** Thi Minh Phuong Ngo, Thanh Hoi Nguyen, Thi Mong Quyen Dang, Thi Van Thanh Do, Alissara Reungsang, Nareekan Chaiwong, Pornchai Rachtanapun

**Affiliations:** 1Department of Chemical Technology and Environment, The University of Danang—University of Technology and Education, Danang 550000, Vietnam; ntmphuong@ute.udn.vn (T.M.P.N.); dtvthanh@ute.udn.vn (T.V.T.D.); 2VN-UK Institute for Research and Executive Education, The University of Danang, Danang 550000, Vietnam; hoi.nguyen@vnuk.edu.vn; 3Department of Postharvest and Food Processing, Faculty of Food Technology, College of Food Industry, Danang 550000, Vietnam; mongquyen76@gmail.com; 4Department of Biotechnology, Faculty of Technology, Khon Kaen University, Khon Kaen 40002, Thailand; alissara@kku.ac.th; 5Research Group for Development of Microbial Hydrogen Production Process, Khon Kaen University, Khon Kaen 40002, Thailand; 6Academy of Science, Royal Society of Thailand, Bangkok 10300, Thailand; 7Materials Science Research Center, Faculty of Science, Chiang Mai University, Chiang Mai 50200, Thailand; meen.nareekan@gmail.com; 8Division of Packaging Technology, Faculty of Agro-Industry, Chiang Mai University, Chiang Mai 50100, Thailand; 9Cluster of Agro Bio-Circular-Green Industry (Agro BCG), Chiang Mai University, Chiang Mai 50100, Thailand; 10Center of Excellence in Materials Science and Technology, Chiang Mai University, Chiang Mai 50200, Thailand

**Keywords:** coating, mango fruit, nanochitosan, pectin, shelf-life, storage temperature

## Abstract

The aim of extending shelf-life and maintaining quality is one of the major issues regarding mango fruit preservation. The quality of mango fruits is greatly affected by postharvest factors, especially temperature and fruit treatment. In this study, the effect of coating and storage temperature on the characteristics of mango fruits was investigated. The mango fruits were immersed in different concentrations (1.5%, 2.0%, and 2.5%) of pectin/nanochitosan dispersion (with ratios of pectin:nanochitosan 50:50), and (0.75%, 1% and 1.25%) of nanochitosan dispersion and stored at 17, 25, and 32 °C for 24 days. Changes in fruit, including weight loss, firmness, color, chemical composition (such as the total soluble solids concentration (TSS)), total sugar, reducing sugar, titratable acidity (TA), and vitamin C were periodically recorded. The results indicated that the pectin/nanochitosan coating significantly prevented reductions in the fruit weight, firmness, TSS, TA, and vitamin C content. Additionally, pectin/nanochitosan at a low temperature (17 °C) had a greater positive effect on fruit shelf-life and weight maintenance than 25 and 32 °C. The coated mango fruits maintained good quality for 24 days at 17 °C, while coated fruits stored at 25 °C and 32 °C, as well as uncoated ones stored at 17 °C, were destroyed after two weeks. At the maximum storage time evaluated, the coating formulations containing pectin and nanochitosan exhibited microbial counts below the storage life limit of 10^6^ CFU/g of fruit. In general, the results showed that the pectin/nanochitosan coating (2%) with a storage temperature of 17 °C is the most effective strategy for improving quality and extending the shelf-life of mango fruits.

## 1. Introduction

Mango is one of the most popular fruits all over the word, due to its attractive taste, odor, and nutritional properties. However, mangoes are climacteric and ripen rapidly after they are harvested, which limits their storage, handling, and transport potential [1]. 

Nowadays, there are many modern methods for fruit preservation, such as modified storage and controlled atmosphere storage, but the application of these methods is limited, due to high technology requirements and costs. Moreover, CO_2_ injury, increased ethanol production, and flavor problems (due to anaerobic respiration) have been reported. The concentration of flavor volatiles and flavor quality were affected by harvest maturity, controlled atmosphere, and storage temperature [2,3]. The application of edible coatings is a new approach to improve the storage life of fruit. Edible coatings are used on fruits to extend shelf-life and improve appearance. Coatings can retard ripening and water loss, as well as reduce decay [4,5]. The atmosphere created by coatings can change in response to environmental conditions, such as temperature and humidity, due to combined effects on fruit respiration and coating permeability [6]. 

Chitosan is one of the most used edible coatings (since it is a natural, biodegradable polymer and non-toxic), proposed as a powerful material that can be applied to engineering, biotechnology, and medicine [7,8,9]. The nanochitosan is considered to have higher antimicrobial activities than chitosan, it is not also harmful for humans [10]. The function of nanochitosan, as an antimicrobial material, is attributed to amino groups or hydrogen bonding between nanochitosan and extracellular polymers [11]. As a biopolymer, nanochitosan has excellent film-forming properties and is able to form a semipermeable film on fruit, which may modify the internal atmosphere, as well as decrease weight loss and shriveling (due to transpiration), thus improving the overall fruit quality. The combination of pectin and nanochitosan is an attractive coating for fruit preservation, due to its antimicrobial and barrier properties [12]. In our previous research, pectin/nanochitosan dispersion was efficiently tested against four popular types of microorganisms, which were *Escherichia coli*, *Saccharomyces cerevisiae*, *Aspergillus niger*, and, especially, *Colletotrichum gloeosporioides* [13]. According to Eshghi et al. [14], a nanochitosan-based coating was successfully applied on strawberries; this coating significantly extended the shelf-life and maintained the bioactive components of the strawberry fruit. It would be of great importance to evaluate the effect of these coatings on the shelf-life of other perishable fruits.

In addition to pectin/nanochitosan coating, temperature management is one of the most important tools for extending the shelf-life of fruits (with the main environmental factor influence on fruit quality) [15]. Due to the short storage life of these fruits, appropriate methods are required to maintain its postharvest quality; increasing the postharvest life can play an important role in delivering mango fruits to distant markets. Therefore, this study aimed to evaluate the effect of pectin/nanochitosan coating and temperature storage on maintaining fruit quality and increasing the shelf-life of mango during storage.

## 2. Materials and Methods

### 2.1. Materials

Chitosan (with molecular weight 70 kDa and degree of acetylation 90%) was purchased from Vietnamese Chitosan company, Kien Giang, Vietnam. Methacrylic acid (MAA), potassium persulfate (K_2_S_2_O_8_), and acetic acid (CH_3_COOH) were purchased from Merck, Darmstadt, Germany. Pectin (P) was extracted from Yanang leaves by Ngo et al. Pectin was extracted using a heating method (with conditions: 88 °C; 75 min; 6.5% citric acid). The DE value of pectin is 48.36 [16]. Glycerol and anhydrous calcium chloride (CaCl_2_) were purchased from Xilong, Science, China. Nanochitosan (NaCS) were prepared as described by Ngo et al. [13]. Firstly, chitosan, with the chitosan concentration of 0.8 wt.%, was dissolved in 0.5 % *v*/*v* MAA solution for 12 h under magnetic stirring. Next, K_2_S_2_O_8_, with the amount of 0.6 mmol, was added to the chitosan/methacrylic solution. After that, the solution was continuously stirred at 70 °C for an hour to form nanochitosan. Then, the mixture was cooled in an ice bath and centrifuged for 30 min at 4000 rpm. Nanochitosan were twice dissolved in distilled water, producing nanochitosan dispersion (NaCS). The structure of the nanochitosan was studied by SEM at 100,000 and 150,000 magnifications, based on our previous research. In this work, the particles were nearly spherical and the particle size diameters of the nanochitosan were less than 100 nm [13]. The nanochitosan dissolved in distilled water was calculated to have 2% nanochitosan dispersion (*w*/*v*).

Mature green mango fruits (*Mangifera indica* L.) (Figure 1), from the Dak Lak province, Vietnam, were harvested and transported to the Lab of the University of Technology and Education, Danang, within 24 h. Uniform mango fruits in color, shape, and weight (390–410 g) were used for this research. They were washed with water and then with ethanol 60% and dried before treatment with the coatings.

### 2.2. Methods

#### 2.2.1. Experimental Treatments

Experimental treatments were conducted at three concentrations of pectin/nanochitosan (1.5%, 2%, and 2.5%), three concentrations of nanochitosan (0.75%, 1%, and 1.25%), and three storage temperature (17, 25 and 32 °C). 

Nanochitosan dispersion was prepared based on the method described by Ngo. et al. [13]. Then, a 2% solution of pectin/nanochitosan was prepared by mixing of water solution of pectin (2%) and nanochitosan (2%) at proportions of 50:50, as following: 2% solution of pectin was prepared by dissolving 2 g of pectin in 98 mL of distilled water at 60 °C. Glycerol was used as a plasticizer at 50% *w*/*w* of polymer. The pH of the pectin solution was adjusted to 4.5 using 2 M Na_2_CO_3_. Then, the mixture of pectin and nanochitosan dispersion was made by slowly adding nanochitosan dispersion into the pectin solution and homogenizing for 1 h with a stirrer. The mixture was stored at 4 °C without mixing for 24 h to degas. 

In the first step of the research process, 126 fruits were separated in seven lots by treatment, with 3 fruits per experiment and subunit for the treatment. The three lots were immersed in three concentrations of 0.75%, 1%, and 1.25% (*w*/*v*) nanochitosan dispersion for 3 min; the three lots were immersed in three concentrations of 1.5%, 2%, and 2.5% pectin/nanochitosan dispersion for 3 min. The rest were immersed in distilled water, as a control. After applying the treatment, the fruits were allowed to dry for 2 h at 25 °C (the average thickness of the coating after drying was about 15 μm by pectin/nanochitosan dispersion with concentration of 2% *w*/*v*). Fruits were then placed onto cardboard trays by hand. All the fruits were stored at 25 °C with relative humidity of 75% for 15 days, and the following experiments were carried out every three days [17]. 

In the second step of the research process, 162 fruits were separated in two groups, for treatment with 3 fruits per experimental. The fruits of a first group were immersed in the best coating solution (chosen from results of first step) for 3 min. The rest was immersed in distilled water, as a control. After applying treatment, fruits were allowed to dry for 2 h at 25 °C. In each group, the mangoes were split three lots. The fruits were then placed onto cardboard trays by hand. The three lots were stored at different temperature 17, 25, and 32 °C with relative humidity of 75% for 24 days. Three fruits were taken from each lot at 3-day intervals and analyzed for fruit quality [17]. 

#### 2.2.2. Determination of Weight Loss 

The fruits were weighed every 3 days during the storage period, using a digital balance (Sartorius, TE313S). The weight loss was determined by the rate of the difference between the initial weight and final weight [5]. 

#### 2.2.3. Determination of Firmness 

Fruit firmness was measured using a TA. XT plus texture analyzer with a 2 mm diameter and 25 mm length puncture probe, calibrated with a 5 kg load cell. Initial grip separation was set at 30 mm, with a test speed of 5 mm/s and a depth of 5 mm. The maximum force (N) was measured at 3 positions—basal, middle, and upper positions of each fruit. The fruit firmness value of each fruit was calculated as mean of 3 measurements. The tests were repeated three times [18].

#### 2.2.4. Determination of Color

The color of the mango skin and mango flesh was measured using a Minolta chroma meter (CR-400, Japan). The tests were in triplicate on three positions of each fruit (basal, middle, and upper) on the skin and flesh of mangoes. The average of the results represented the color value. The results were determined in the color spaces L*, a*, b*, chroma, and hue. The color characteristics of mango peel and flesh were obtained using a Konica Minolta. L* measures lightness and varies from 100 (for perfectly reflective white) to 0 for (perfectly absorptive black); a* measures redness when positive, gray when zero, and greenness when negative; and b* measures yellowness when positive, gray when zero, and blueness when negative. Hue angle calculation: h° = arctan (b*/a*). The Minolta chroma meter was calibrated with a white standard tile: L = 97.62, a = −0.01, b = 1.56, C = 7.73, and hue = 91.39 [19]. 

#### 2.2.5. Determination of Chemical Characteristics: Total Soluble Solids (TSS), Total Cidity (TA), Reducing Sugar and Vitamin C

Thirty grams of mango flesh were homogenized with a stirrer in 150 mL of distilled water for 2 min and then filtered [20]. 

Total soluble solids (TSS) concentration was determined using a digital refractometer PR-101*α* (Atago Co., Ltd., Kobe, Japan) at 25 °C and expressed as % of dry matter [21] (see Appendix A). 

Total acidity (TA) was measured using 10 mL of the above mixture by titration, with 0.1 N NaOH up to pH 8.1. The results were expressed as g citric acid equivalent per 100 g fresh weight [20]. 

Vitamin C (Ascorbic acid) content was determined by colorimetry using 2,6-dichlorophenolindorhenol titration (according to AOAC method, 1984) [21]. 

#### 2.2.6. Determination of Microbial Analysis 

The total microbial count was determined by the plate count agar method, according to ISO 4833, standards 8552 and 17410 [22]; the count of the cells of yeasts and molds was determined according to AOAC 997.02 [23]:

The change in microbiological analysis of the fruit during storage at different storage temperatures was determined as follows: 20 g mango fruit was dissolved into sterile containers, then 180 mL of peptone solution 0.1% was added, and the mixture was homogenized with a blender for 2 min. A total of 0.1 mL of the prepared solution was poured on Petri dishes containing PDA (potato-dextrose agar) medium. This was incubated at 30 °C for 24 h. The number of yeasts and mold cells were counted [22,24].

Similarly, 0.1 mL of the solution prepared as above was poured into agar plates (prepared according to ISO 4833, standards 8552 and 17410) and incubated at 30 °C for 48 h. Then, the number of colonies forming units (CFU) per unit mass of 1 g fruit were counted and calculated [23,24].

#### 2.2.7. Statistical Analysis 

The experiment was performed in a completely randomized design (CRD), with three repetitions. All data were analyzed by one-way ANOVA. Mean separation was performed by Duncan’s multiple range tests, with significance level (*p* ≤ 0.05). Statistical analysis was carried out by using Minitab16 software (Version 8.0, Northampton, UK) [25].

## 3. Results and Discussion

### 3.1. Effect of Coating on Shelf-Life and Some Quality Traits of Mango Fruits

#### 3.1.1. Weight Loss 

The weight loss of the mango fruits increased progressively in all treatments during storage (Figure 2). The weight loss of fruits depends on coating. The weight loss of control sample (uncoated) was 21.8% after 9 days of storage, whereas the weight losses for mangoes coated with 1.5%, 2%, and 2.5% pectin/nanochitosan were 9.34%, 7.07%, and 6.43%, respectively; coated with 0.75%, 1%, and 1.25 % nanochitosan were 12.87%, 10.25%, and 8.97%, respectively. 

Based on the results, the pectin/nanochitosan-coated fruits (2% and 2.5%) presented the lower weight loss than the others after 15 days of storage at 25 °C. Unlike the control and 0.75%, 1% nanochitosan and 1.5% pectin/nanochitosan, higher nanochitosan concentration (1.25%) and higher pectin/nanochitosan concentration (2% and 2.5%) led to an increase in the storage life of the mango fruit (up to 15 days). The weight losses for mangoes coated with 2% and 2.5% pectin/nanochitosan were 12.5% and 11.2%, respectively, and those coated with 1.25% nanochitosan were 15.79%, whereas the mango coated with 1.25% pectin/nanochitosan, 0.75% and 1% nanochitosan were spoiled after 12 days of storage. Fruits without coating (control) showed the highest weight loss. These results showed that coating significantly maintained fruit weight and this effect was significantly higher at 1.25% nanochitosan and 2%, 2.5% pectin/nanochitosan. The reason for the difference in weight loss among samples might be due to the coating and coating thickness, which affected the transpiration and respiration rates of the mango. In our previous research, the water vapor transmission rate of pectin/nanochitosan films (47.7 μm in thickness) and nanochitosan films (43 μm in thickness) were 8.1 and 9.24 g/m^2^/day, respectively; the oxygen transmission rate of pectin/nanochitosan and nanochitosan films were 18.63 and 320.8 cc/m^2^/day, indicating that pectin/nanochitosan films were better barriers, compared with nanochitosan films [13]. These results were consistent with previous studies on apple fruit (Ali Sahraei Khosh Gardesh et al., 2016) [26]. Eshghi et al. [14] also reported that nanochitosan-based coating significantly reduced the weight loss of the strawberry fruit. 

#### 3.1.2. Firmness 

Firmness is an important factor affecting on the transportation of mango fruits, especially for the export of mango fruits [27]. The changes in firmness of all samples are shown in Figure 3. 

According to the results, the decrease of firmness during storage time was observed for all treatments. For the control (uncoated), the firmness decreased rapidly from the initial value of 31.22 N to 14.16 N and 0.88 N after 3 and 9 days of storage, respectively. Mango fruits coated with pectin/nanochitosan or nanochitosan significantly delayed loss of firmness. The higher the coating concentration applied, the slower the mango firmness decreased. The firmness of mango samples coated with 1.5%, 2%, and 2.5% pectin/nanochitosan after 9 days of storage was 6.4, 15.44, and 19.87 N, respectively. Fruit texture is affected by cell turgidity and the structure and composition of the cell wall polysaccharides [28]. Maintaining the fruit firmness could probably be explained by the fact that the coating inhibited moisture loss and delayed the degradation of insoluble protopectins to soluble pectin and pectic acid [29], as well as other reactions by oxygen. During fruit ripening, de polymerization or shortening of chain length of pectin substances occurs with an increase in pectin-esterase and polygalacturonase activities [30]. Additionally, nanochitosan coating or pectin/nanochitosan coating led to reduce oxygen availability, which reduced these enzymes and increased retention of fruit firmness during storage [31].

In the findings, the weight loss and firmness of the mango depended on the coating. The weight loss of mangoes increased and mango firmness reduced slowly when increasing pectin/nanochitosan or nanochitosan concentration. However, pectin/nanochitosan coated fruits (2.5%) were not ripening so as to make it soft enough to eat. Therefore, mangoes coated with pectin/nanochitosan-coated fruits (2.0%) exhibited less fruit loss and permitted the ripening process to continue during storage time. 

#### 3.1.3. Color 

Figure 4 shows that the L* value of the mango peel in all samples increased during storage time, except for the samples coated with 2.5% pectin/nanochitosan. 

The coating had an effect on delaying color changes of mango peel, in comparison to the control samples (uncoated). After nine days of storage, the L* value of the control mango peel was 72.1, while the L* values of mango peel coated with the 1.5%, 2%, and 2.5% pectin/nanochitosan were 58.7, 53.5, and 50.4, respectively. A slow increase in L* value indicates a slow color change. However, the L* value of mango peel coated 2.5% pectin/nanochitosan decreased and started decreasing sharply after 12 days of storage. This is probably ascribed to the browning of the mango. Similarly, L* value of the mango flesh samples increased in the first days and then gradually decreased. L* value of mango flesh of the control samples and coated mango sample (except for coated with 2% pectin/nanochitosan) decreased constantly after 3, 6, and 9 days of storage, which indicated that the mangoes ripened and began to spoil. The L* value of the mango flesh of the sample coated with 2% pectin/nanochitosan increased slowly during 15 days of storage. The 2.5% pectin/nanochitosan coatings had a negative effect on the mango color, namely the L* value decreased significantly at the end of the storage time. This can be probably explained by the fact that the thick coating reduces the amount of diffusive oxygen into the mango, which plays an important role in delaying the respiration process. This relates to changes in lightness of the peel and flesh.

Changes in the color of the peel and flesh result from both chlorophyll degradation and carotenoid synthesis during mango ripening, which turns peel color from green to yellow with ripening of mango fruits. The change of a*, b*, and hue angle values indicates that coated fruits ripen slower than uncoated fruits. Changes in a* and b* values of peel are also shown in Figure 4. As the results, the a* value was more negative in coated samples, indicating the mango peel to be greener. The storage time related color shift towards positive a* value indicates more redness in color that is the result of ripening. Figure 4 shows an increase in b* value of mango peel, with highly significant difference between coated and uncoated samples. However, b* value peel of mango peel—coated with 2.5% pectin/nanochitosan decreased significantly after 15 days. This decrease in b* value indicated an increase toward darker chroma. Changes in the a* value showed a decreasing trend; the b* value and hue* angle showed an increasing trend for the control and coated fruits in mango flesh during storage. After 15 days of storage, the time related changes in b* values for the coated samples started to show significant differences from the control. The lowering decrease of b* value indicates a remain in yellowness of samples. 

Hue angle values of both peel and flesh of mango coated with pectin/nanochitosan were higher than ones coated with nanochitosan. It means mango fruits coated can be probably delayed the ripening better than ones coated with nanochitosan. After 15 days of storage, the hue values of the mango peel, coated with 1.25% nanochitosan, 2%, and 2.5% pectin/nanochitsoan, were 74.1, 86.78, and 56.75, while of flesh were 80.5, 92.15, and 75.68, respectively. At the end of storage time when mango fruits were cut out, browning was noticeable for coated and uncoated mango fruits except for mangoes—coated with 2% pectin/nanochitosan. This indicated that all samples were spoiled after 15 days storage at 25 °C.

Regarding the color of mango peel and flesh during storage, 2% pectin/nanochitosan was the appropriate coating because the color changes of both mango peel and flesh were the slowest and 2.5% pectin/nanochitosan also delayed the change in color of mango but browning occurred, which had a negative effect on the sensory quality. The actual colors of uncoated and all coated mangoes were also presented in Figure 5.

#### 3.1.4. Chemical Characteristics

Total soluble solid values of mango fruits, in all treatments, are shown in Figure 6. 

At first, the total soluble solids of mango fruit showed an increasing trend and then a decreasing trend. Both the nanochitosan coating and pectin/nanochitosan coating delayed the increase of total soluble solids of mangoes. The higher concentration of coating dispersion showed the slowest increase of total soluble solid. However, the TSS of mango fruits coated with 2.5% pectin/nanochitosan increased slowly during fruit ripening in storage. This was consistent with the results of fruit firmness. These results indicates that mango fruits coated with 2.5% pectin/nanochitosan were not able to continue ripening. A similar result was observed when the chitosan coating was applied for some fruits according to Bautista-Banos et al. [32] or in another research, chitosan coating decreased the TSS of fruit juice in mango [33,34,35]. Additionally, based on the results, the total acid and vitamin C of mango fruit generally decreased during storage. At first, it showed a slow decreasing trend and then a significant decreasing trend. Both the nanochitosan and pectin/nanochitosan coating treatment delayed the decrease in content of total acidity and vitamin C. Similar to TSS, the total acidity of mango fruits coated with 2.5% pectin/nanochitsan was higher than others; this could be explained by the fact that mango fruits could not continue ripening and the respiration rate may be inhibited by coating.

Conclusion: Above results showed that mangoes coated with 2% pectin/nanochitsoan decreased the weight loss, delayed the respiration process, and allowed the ripening process. The mango fruits coated with 2.5% pectin/nanochitsoan were browned after 15 days of storage, and they were unable to continue ripening. These results caused low oxygen and carbon dioxide permeability of coating and resulted in a sudden inhibition in the ripening process and inhibited the normal respiration of the mangoes. The effectiveness of 1.5% pectin/nanochitsoan coating or 0.75%, 1%, 1.25% nanochitosan coating was lower than 2% pectin/nanochitsoan coating. This is probably due to the reason that the blending of pectin with nanochitosan led to an improvement of oxygen and water vapor barrier properties of these films, compared to nanochitosan films solely according to our previous researches [13]. These results were consistent with the report of Silva et al. who indicated that mango preserved with 1%, 2%, and 3% of chitosan solution delayed the change in physical and chemical characteristics, and 2% of chitosan solution showed the least color changes [12]. Therefore, 2% pectin/nanochitsoan coating were chosen for the further research.

### 3.2. Effects of Storage Temperature on Shelf-Life and Some Quality Traits of Mango Fruits

Mango fruits were immersed in the 2% P/NaCS solution for 3 min and then dried for 2 h at 25 °C. The effect of temperatures was studied at different storage temperatures of 17 °C, 25 °C, and 32 °C. Besides, the effect of temperatures without considering coating effect was also studied fruits stored at different storage temperatures of 17 °C, 25 °C, and 32 °C.

#### 3.2.1. Weight Loss 

The effects of storage temperature and pectin/nanochitosan coating were significant on fruit weight loss (Figure 7). The results showed that low storage temperature significantly reduced weight loss of mango fruits over the 24 days of storage. The weight losses of mango fruits were in the range of 14.3–26.7%, depending on coating and storage temperature. The weight loss of the coated fruits was lower than that of the uncoated fruits at all temperatures. After 9 days of storage, weight loss of the uncoated samples was 23.2, 21.8, and 15.8%, while the weight loss of the coated samples was 12.6, 8.5 and 5% at 32, 25, and 17 °C, respectively.

It is said that the allowable limit of fruit weight loss during storage is about 15% [10]. In the present study, application of coating was significantly effective for maintaining the weight loss The longer the storage duration, the larger the difference in weight loss between the coated and uncoated mango fruits. The weight loss of mango fruits during storage is probably related to physiological processes, such as respiration and evaporation, and other biological processes of mangoes. Coating reduces the weight loss of mangoes because the coating acts as a semi-permeable layer, which can inhibit gas and moisture exchanges, so reducing respiration and transpiration rates and the oxidation process. Our results are consistent with Ali et al. who reported that the weight loss of tomato can be reduced by coating with Arabic gum [36]. Another study also reported that plums coated with corn starch have minimal weight loss, which is explained by the modification of gas composition. Our results also showed that decreasing temperature decreased fruit weight loss. This could be explained by the fact that low temperature reduced the biochemical reactions and water loss rate, which causes the weight loss.

Regarding the weight loss, the longest storage time of uncoated mangoes was 15 days at 17 °C, while the longest storage times of coated mangoes were 24 days at 17 °C and 15 days at 25 °C. 

#### 3.2.2. Firmness

Figure 8 shows that the firmness of coated and uncoated mangoes decreased during storage time. 

Pectin/nanochitosan coating and low storage temperature had a positive impact on firmness maintenance of mango fruits. The firmness change rate of uncoated mangoes at 17 °C was approximately equal to that of coated mangoes at 32 °C. This indicated that coating is significant for firmness maintenance. These results also showed that firmness maintenance at storage temperatures of 17 ℃ and 25 °C is much better than those at 32 °C. Interestingly, mango fruits coated with 2% pectin-nanochitosan could be able to continue ripening at storage temperatures of 17 °C; both of uncoated and coated mangoes reached a moderate softness of 2 N to 6 N, ready for consumption. Mango fruits coated with pectin/nanochitosan could be attributed to O_2_ exchange barrier, thereby reducing respiration and delaying converting protopectin to soluble pectin and conversion of starch to glucose, which resulted in maintaining a better firmness during storage. The current study also confirms that fruits stored at the cold storage condition had better fruit firmness and acceptable ripening quality during low temperature storage (17 °C), compared to storage at room temperature (25 °C and 32 °C). Low temperature reduces respiration and metabolic processes thereby slowing down losses in the rate of fruit firmness during fruit storage [37]. The results indicated that the coatings and storage temperature positively impacted on the retention of firmness.

In our findings, in terms of acceptable weight loss and firmness for consumption of mangoes, the longest storage time for uncoated mango was 15 days at 17 °C, while coated mango lasted for storage times of 12, 15, and 24 days at 32 °C, 25 °C, and 17 °C, respectively.

#### 3.2.3. Color

Effect of coating and storage temperature on color of mango fruits affected by coating and storage temperature are shown in Figure 9. Color was evaluated based on the lightness (L* value) and Hue angle. 

Figure 9 shows that the L* values of mango peel increased during storage time. The L* value of the control samples had a slower increase at 17 °C, compared to those at 25 °C and 32 °C, but the L* value decreased after 9 days of storage. The increase in L* value of coated mango samples was slower than that of uncoated mango samples at all storage temperatures. When the storage temperature decreased, the L* value of the coated mango samples increased slowly. Lowering the storage temperature of uncoated mango samples delayed the ripening process, which kept the mango green and resulted in a low lightness. Normally, ripe fruits turning yellow increases the lightness of fruits. The L* value of the mango flesh was similar to the L* value of the mango peel. However, when mangoes were ripe, the flesh turned dark yellow, which explains the reason that the L* values of all mango samples decreased at the end of the storage time. The lower L value changes in the coated mango may be related to the effect of the coating in creating modified atmospheres within the mango fruit. Contrast to L value, hue angle of mango fruits decreased during fruit ripening in storage. Hue angle of mango fruit also decreased with increasing storage temperature. This indicated that lower storage temperature slowed down the de-greening processes in mango fruits. These results were consistent with some previous researches in apricots and peaches [38]. The actual colors of uncoated and coated mangoes were also presented in Figure 10.

#### 3.2.4. Chemical Characteristics 

The effects of storage temperature on total soluble solids (TSS), titratable acidity (TA), and vitamin C contents of mango fruits are shown in Figure 11.

According to Medlicott and Thompson [39], the major sugars of mangoes were identified as glucose, fructose, and sucrose. Sucrose was found to be in the greatest concentration throughout, with fructose the predominant reducing sugar. Our results were consistent with findings of Medlicott and Thompson [39]. TSS steadily increased during storage time. TSS of the control mango was higher than coated mangoes. A slower increase in TS was observed at a lower temperature. TSS content of 20% was the acceptable level during the ripening process. TSS of control samples declined at the end of the storage time, while TSS of coated samples remained at 20% at all storage temperatures.

Decrease of total acidity in mangoes is caused by a decrease of citric acid and malic acid [40]. There was a slower decrease in titratable acidity of the coated mangoes, compared to the control mangoes at all storage temperatures. Vitamin C content of the coated samples was in the allowable range of 29.1 to 5.6 mg/100 g, while vitamin C content of the control samples was lower than this range. Vitamin C content of mango fruits coated with pectin/nanochitosan was higher, in comparison to the control samples. The lower the storage temperature was, the slower the degradation rate of titratable acidity and vitamin C was. Maintenance of total acidity content and ascorbic acid of coated fruits has been reported by other researchers [40]. 

In our recent finding, a decrease of TSS contents and the maintenance of total acidity and vitamin C content could be explained by the fact that the pectin/nanochitosan coating could form a thin film with micropores and delay the physiological and biochemical processes, such as the conversion of starch into sugars, by modifying the internal atmosphere of the package [41]. At the end of the storage time, the control sample began deteriorating, due to increasing respiration rate, thereby reducing the chemical compositions content. In addition, TSS contents had a linear correlation with the mango firmness and color during storage time. All coated mango samples maintained TSS and vitamin C contents in the acceptable limit range for 12, 15, and 24 days, at storage temperatures of 32 °C, 25 °C, and 17 °C, respectively; uncoated mango samples exceeded the acceptable limit range after nine days of storage. 

#### 3.2.5. Microbiological Analysis

Pectin/nanochitosan coating was effective in reducing the number of microorganisms in mango fruits (Figure 12). In previous publications, the microbiological limit for the preservation of fruits and vegetables was 10^6^ CFU/g [42]. The pectin/nanochitosan coatings could maintain the number of total microbials, below acceptable limit of 10^6^ CFU/g fruit, even after 24 days of storage.

The lower the storage temperature is, the smaller the number of microbes is. These results showed that yeasts, molds, and bacteria were present in the mango fruits. The reduction in microbial counts on the surface of coated mango samples is due to the antimicrobial activity of nanochitosan. Furthermore, pectin/nanochitosan coatings provide a partial barrier to oxygen (O_2_), creating a suitable atmosphere environment to prevent or restrict the microbial growth of fruits. The resonant combination of coating and low temperature was highly effective in delaying microbial growth. 

According to some previous research, the application of chitosan could be beneficial in prolonging shelf-life and maintaining the quality of mango fruits by inhibiting the respiration rate in mango fruits and decreasing postharvest diseases [12,43]. In this study, the pectin/nanochitosan not only maintains weight loss and firmness but also improves the postharvest quality during storage; this suggests that pectin/nanochitosan is promising as a coating with higher antimicrobial activity and can be used in commercial postharvest applications for extending the shelf-life of the mango fruits.

## 4. Conclusions

In conclusion, nanochitosan and pectin/nanochitosan coatings are able to maintain the quality of mango fruit to increase its storage time. This research demonstrated that the pectin/nanochitosan coating had a better positive effect than nanochitosan coating on mango preservation. The pectin/nanochitosan coating increases the shelf-life of mango fruit by reducing water loss. It also maintains the firmness and the quality of mango fruits by preventing the reduction of color, and total soluble solids. The positive effect of pectin/nanochitosan on the postharvest quality of mango fruits was enhanced with a concentration of 2% and used along with a low storage temperature. This study found that a storage temperature of 17 °C is the best way to increase the shelf-life and improve the postharvest quality of mango fruits. Changes to the physicochemical and microbiological characteristics of coated mangoes reduced at a storage temperature of 17 °C. Moreover, the coated mangoes showed a decrease of total microorganism in the mango fruits, not exceeding of 10^6^ CFU/g during the storage time. The mango fruits were well-stored under these conditions for more than 24 days; determining other quality traits of mango fruits coated with pectin/nanochitosan requires further studies.

## Figures and Tables

**Figure 1 polymers-13-03430-f001:**
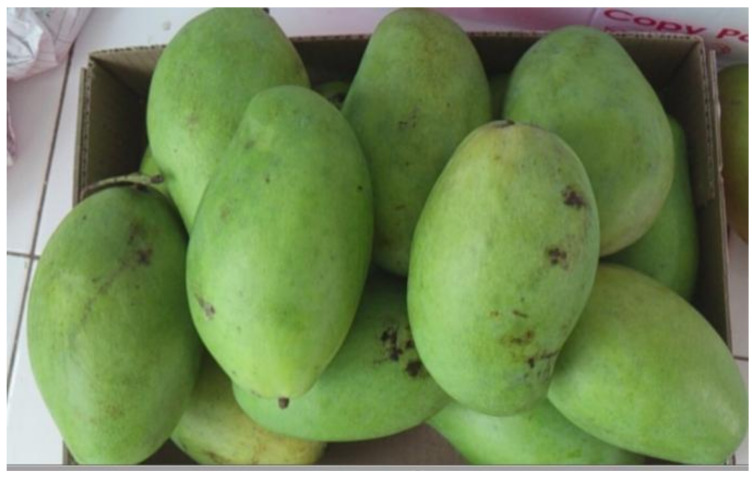
“Elephant” Mangoes.

**Figure 2 polymers-13-03430-f002:**
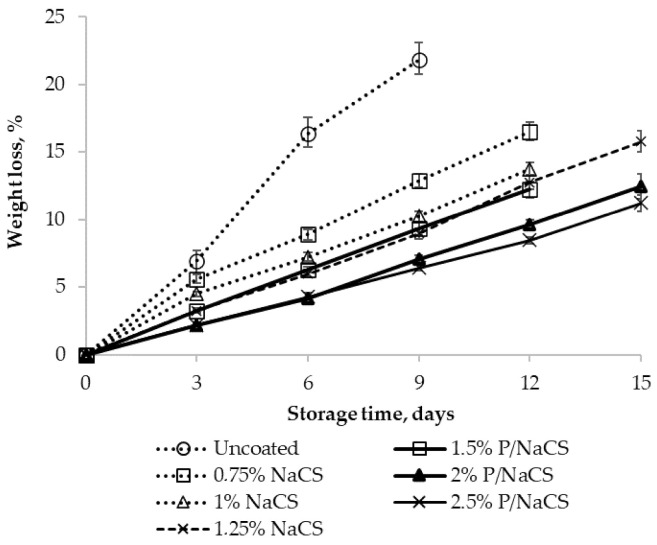
Effect of coating on the weight loss of mango fruits. Numbers are the mean ± standard errors of three replications. Note: P/NaCS means pectin plus nanochitosan with ratio 50:50 for all Figures.

**Figure 3 polymers-13-03430-f003:**
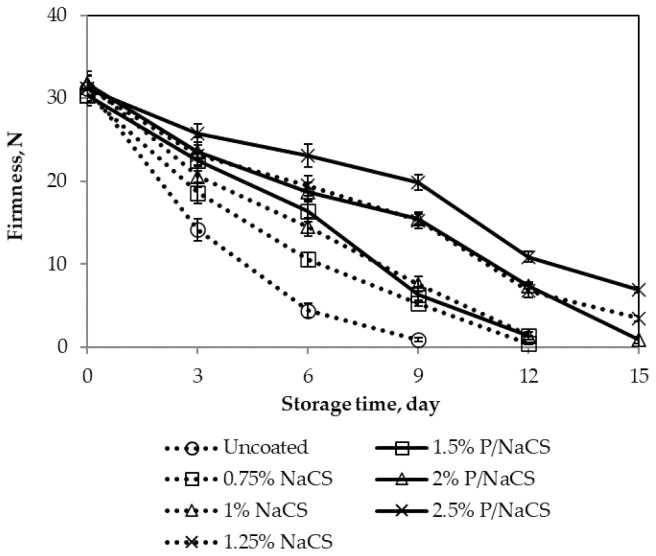
Effect of coating on firmness of mango fruits. Numbers are the mean ± standard errors of three replications.

**Figure 4 polymers-13-03430-f004:**
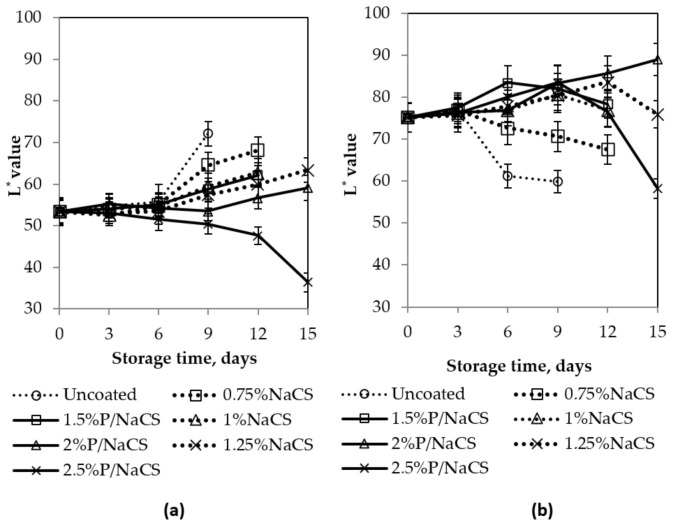
Effect of coating on color of mango fruits. Numbers are the mean ± standard errors of three replications. (**a**) L* value, (**c**) a* value, (**e**) b* value, (**g**) hue angle in mango peel, (**b**) L* value (**d**) a* value, (**f**) b* value, and (**h**) hue angle in mango flesh.

**Figure 5 polymers-13-03430-f005:**
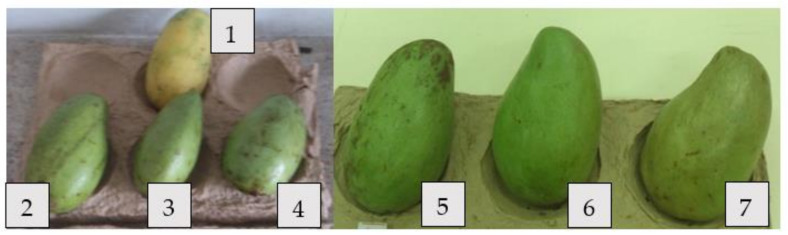
The color of mangoes at 9th day storage: (**1**) the control sample (uncoated); (**2**) 0.75% NaCS coating; (**3**) 1% NaCS coating; (**4**) 1.25% NaCS coating; (**5**) 1.5%P/NaCS coating; (**6**) 2% P/NaCS coating; and (**7**) 2.5% P/NaCS coating.

**Figure 6 polymers-13-03430-f006:**
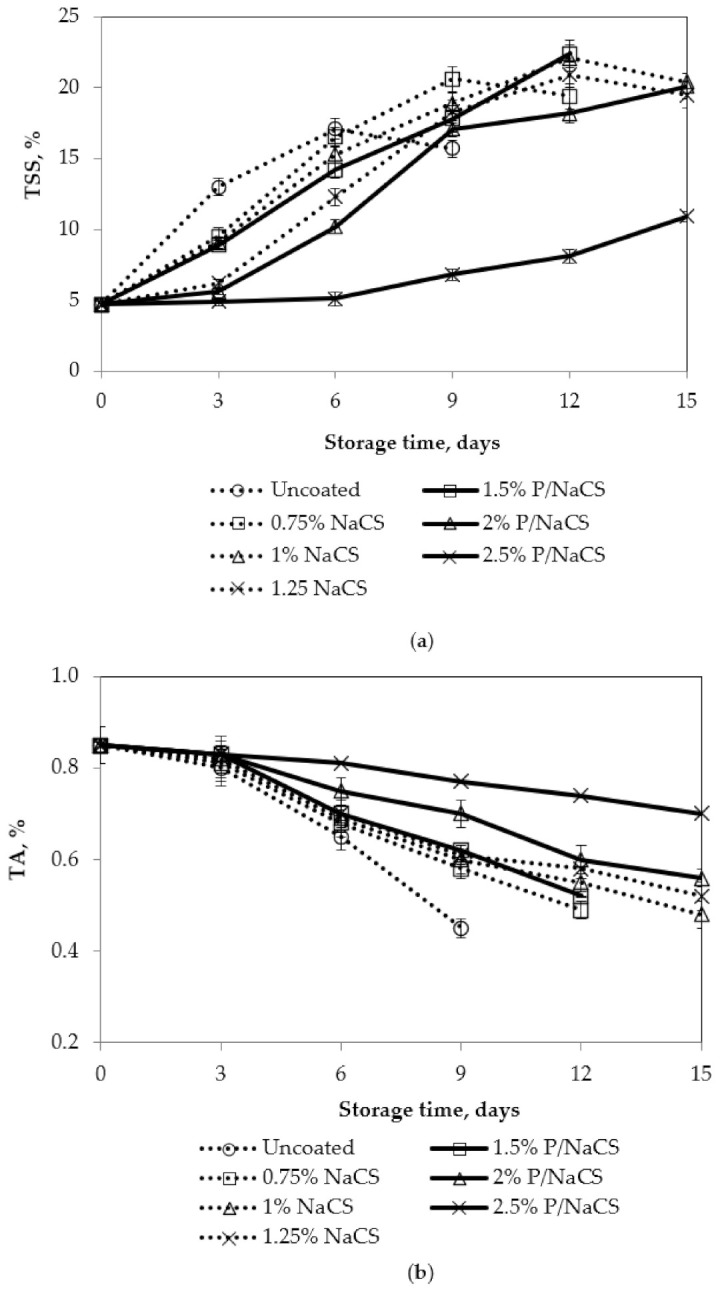
Effect of coating on chemical characteristics of mango fruits. Numbers are the mean ± standard errors of three replications. (**a**) TSS, (**b**) TA, and (**c**) vitamin C.

**Figure 7 polymers-13-03430-f007:**
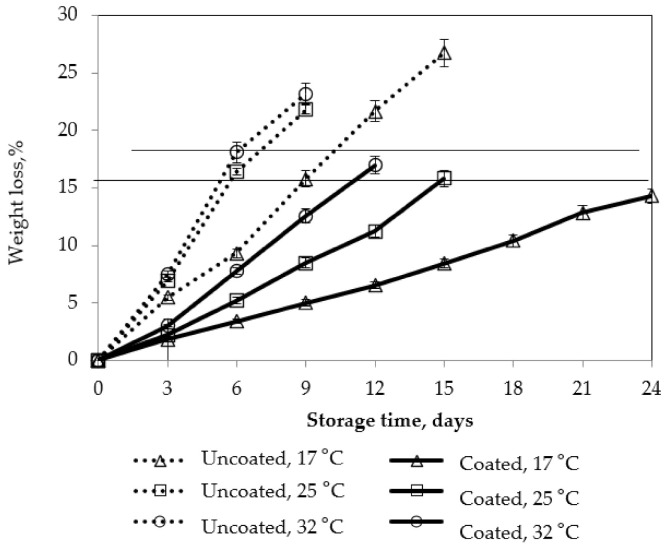
Effect of storage temperature on weight loss of mango fruits. Numbers are the mean ± standard errors of three replications.

**Figure 8 polymers-13-03430-f008:**
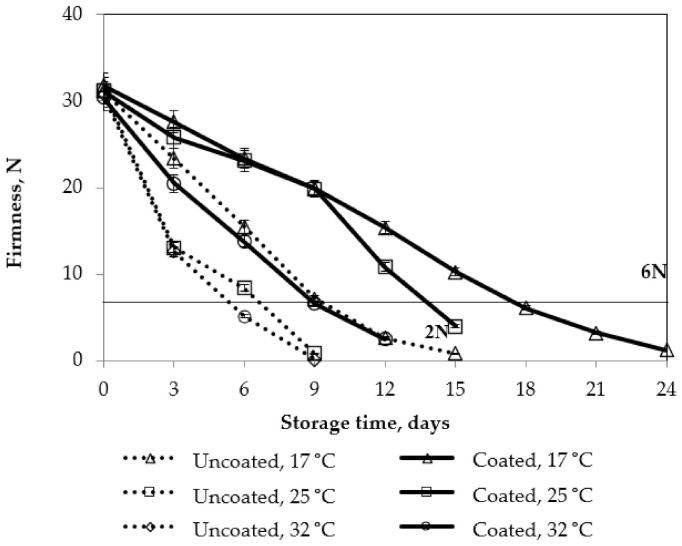
Effect of storage temperature on firmness of mango fruits. Numbers are the mean ± standard errors of three replications.

**Figure 9 polymers-13-03430-f009:**
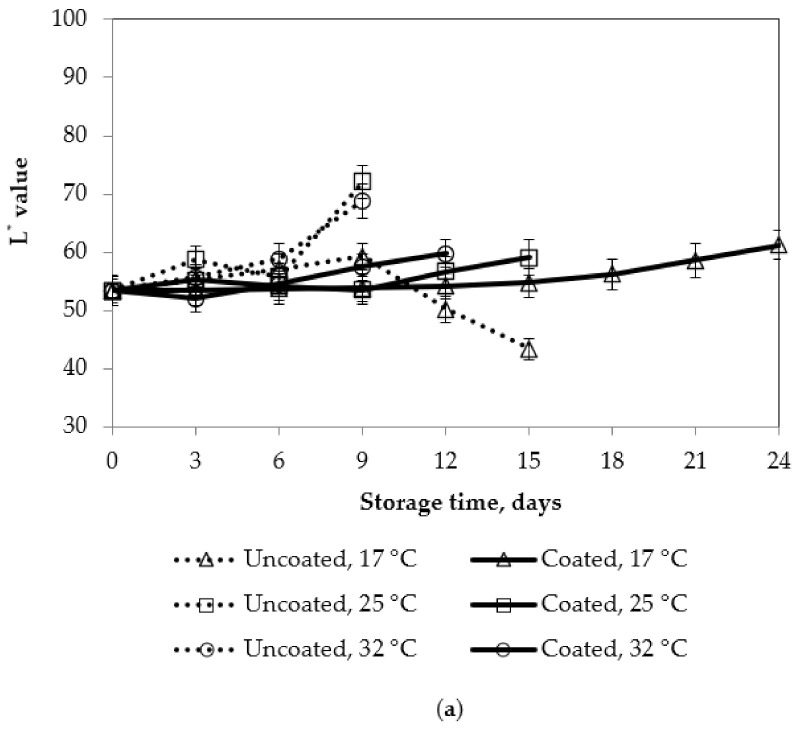
Effect of storage temperature on color of mango fruits. Numbers are the mean ± standard errors of three replications. (**a**) L* value, and (**c**) Hue angle in mango peel, (**b**) L* value, and (**d**) Hue angle in mango flesh.

**Figure 10 polymers-13-03430-f010:**
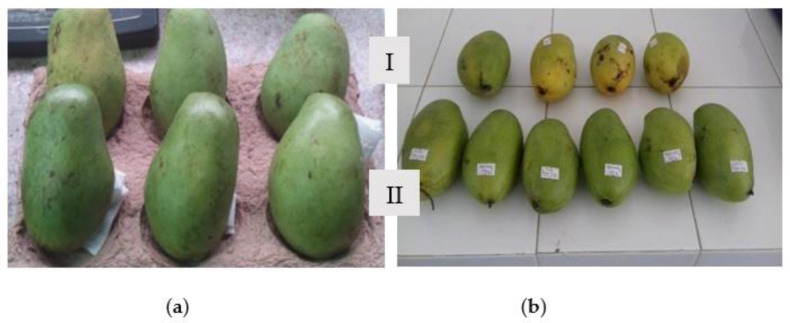
The color of mangoes at 0th day storage (**a**) and 9th day storage (**b**) at 32 °C: Inner row (I): the control sample (un-coated); outer row (II): 2% P/NaCS coating.

**Figure 11 polymers-13-03430-f011:**
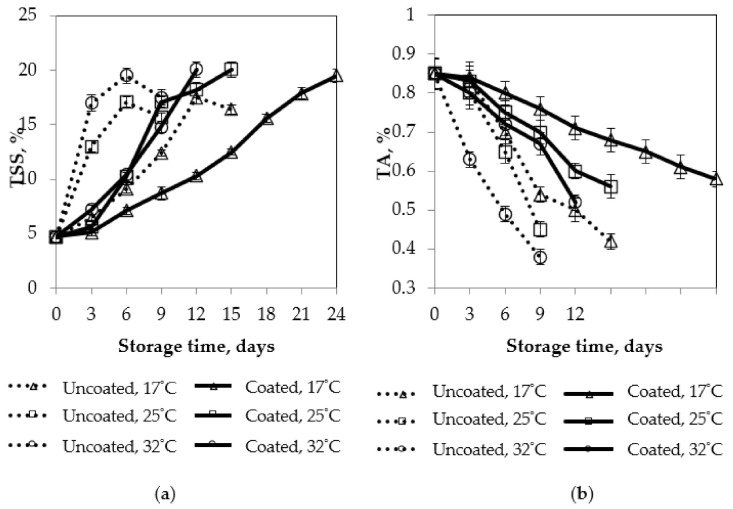
Effect of storage temperature on chemical characteristics of mango fruits. Numbers are the mean ± standard errors of three replications. (**a**) TSS, (**b**) TA, and (**c**) vitamin C.

**Figure 12 polymers-13-03430-f012:**
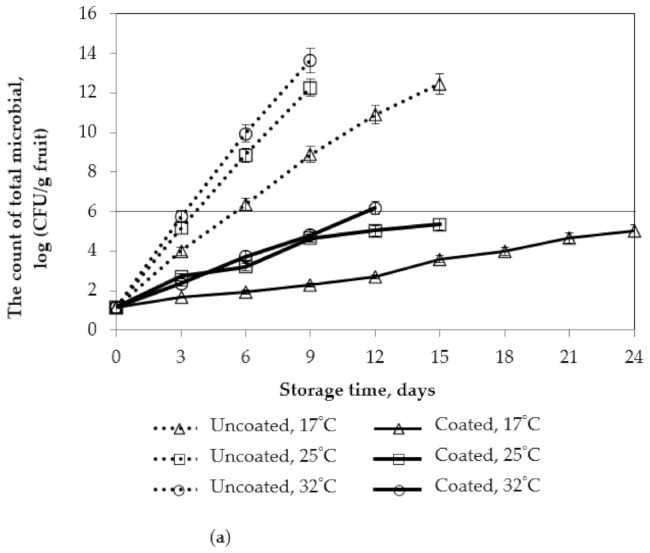
Effect of storage temperature on microbiological analysis of mango fruits. Numbers are the mean ± standard errors of three replications: (**a**) total microbial and (**b**) yeasts and molds.

## Data Availability

The data presented in this study are available on request from the corresponding author.

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
