# Peer review of "Effect of Pectin/Nanochitosan-Based Coatings and Storage Temperature on Shelf-Life Extension of “Elephant” Mango (Mangifera indica L.) Fruit"

_polymers, 2021, doi:10.3390/polym13193430_

Round 1

Reviewer 1 Report

Dear Authors,

This is a paper devoted to studying the effect of different postharvest conservation treatments, based on pectin and nanochitosan, and different storage temperatures on mango shelflife. Several matters will need to be addressed before being considered for publication.

Absract section. .. More quantitative data must be included.

Introduction section.

line 53...define CA

More references must be included in the introduction, they are missing in some parts of it.

Material section.

Please separate the reagents part from methodologies.

Provide the city for each company mentioned in the text.

Line 89...Provide a short description of the pectin isolation method.

Line 98... How was determine the size of chitosan nanoparticles?. Is there any information about the zeta potential value?, please provide it if available.

Line 149...automatic titration ..which equipment was used?

line 169.... Which statistical test did you use for mean statistical comparison?

It is not clear the experimental design used for this work. How many replicates per treatment were used. How many mangoes per treatment were used? How many mangoes were sampled for analysis each time?

Results

It is necessary to provide more information about chitosan nanoparticle characterization. (TEM or SEM, FTIR, TGA,etc) and pectin/nanochitosan nanocomposite characterization. In case this work was made using a previously characterized material, already published on a previous paper, please specify in the text body and provide adequate information.

There is a lack of statistical test analysis in all measurements.

Photography of the treated mangoes at different postharvest times would be interesting to include.

Conclusion

Line 462...especially...grammar mistake

Author Response

The authors thank the editor and the reviewers for reviewing this manuscript and for your comments and suggestions, which greatly improved the quality of the paper. Detailed responses to the comments are provided below.

Reviewer 1 Comments

The authors thank the reviewer for taking the time to review the manuscript. Detailed responses to the comments are provided below.

Abstract section

Question 1: More quantitative data must be included.

Answer: More quantitative data was added in the abstract: “At maximum storage time evaluated, coating formulations containing pectin and nanochitosan exhibited microbial counts below the storage life limit of 106 CFU/g of fruit”. It reads now on page 1, lines 42-44. It was marked in red.

Introduction section.

Question 2:

line 53...define CA

Answer: It was changed. It reads now on page 2, line 58-59. It was marked in red.

Question 3:

More references must be included in the introduction, they are missing in some parts of it.

Answer: More references were added in the introduction. It reads now on page 2, line 67-68. They were marked in red.

Material section.

Question 4: Please separate the reagents part from methodologies.

Answer: Thank you for your reminder. I already correct the whole part 2.2

Question 5: Provide the city for each company mentioned in the text.

Answer: The city of each company is added. It reads now on page 2, line 93, 95, and 98. They were marked in red.

Question 6: Line 89...Provide a short description of the pectin isolation method.

Answer: In my previous study, pectin was extracted from yanang leaves using heating method with conditions (88 oC; 75 minutes; 6.5% citric acid). DE value of pectin is 48.36. This description was added in 2.1 section. It reads now on page 2, line 96-97. It was marked in red.

Question 7: Line 98... How was determine the size of chitosan nanoparticles? Is there any information about the zeta potential value? please provide it if available.

Answer: The size of chitosan nanoparticles was determined by SEM images as below answer. The information of the size of chitosan nanoparticles was added in the manuscript. It reads now on page 3, line 105-108. It was marked in red.

In the research of our colleague about chitosan nanoparticles using the same method, zeta potential value of chitosan nanoparticles was presented in following Table:

Size of nanochitosan, nm

zeta potential value, mV

Concentration of K2S2O8, mmol

This is the link related to the zeta potential value:

 https://www.academia.edu/29741896/NGHI%C3%8AN_C%E1%BB%A8U_T%E1%BB%94NG_H%E1%BB%A2P_NANO_CHITOSAN_B%E1%BA%B0NG_S%E1%BB%AC_D%E1%BB%A4NG_AXIT_METHACRYLIC_V%C3%80_%C4%90%C3%81NH_GI%C3%81_KH%E1%BA%A2_N%C4%82NG_KH%C3%81NG_N%E1%BA%A4M_COLLETOTRICHUM_MUSAE_PH%C3%82N_L%E1%BA%ACP_T%E1%BB%AA_QU%E1%BA%A2_CHU%E1%BB%90I

Question 8: Line 149...automatic titration ..which equipment was used?

Answer: There is a mistake in describing the method. It was rewritten. It reads now on page 4, line 184. It was marked in red.

Question 9: line 169.... Which statistical test did you use for mean statistical comparison?

Answer: CFU counting varies from 10m to 10n, with n and m being different natural numbers. We used log transformation prior analyzing data. We compared the results based on the mean of three replicates and the standard error of the means.

Statistical analysis

The experiment was performed in a completely randomized design (CRD), with three repetitions. Statistical analysis was carried out by using Minitab16 software (Version 8.0, Northampton, USA)

The statistical analysis results were shown in Data availability statement file.

Question 10: It is not clear the experimental design used for this work. How many replicates per treatment were used. How many mangoes per treatment were used? How many mangoes were sampled for analysis each time?

Answer: Thank you for your suggestion. All treatments were performed in triplicate. Three mangoes per treatment were used. Three mangoes were sampled for analysis each time. The experimental design is rewritten. It reads now on page 3-4, lines 133-150. It was marked in red. 

Results

Question 11: It is necessary to provide more information about chitosan nanoparticle characterization. (TEM or SEM, FTIR, TGA,etc) and pectin/nanochitosan nanocomposite characterization. In case this work was made using a previously characterized material, already published on a previous paper, please specify in the text body and provide adequate information.

Answer: Thank you for your suggestion.

- Some information about chitosan nanoparticle characterization was added in the manuscript (2.1 Materials section). It reads now on page 3, lines 105-108. It was marked in red. 

- Some information of pectin/nanochitosan and nanochitosan films was added in Results Section. It reads now on page 5, lines 233-237. It was marked in red. 

 Some information about chitosan nanoparticle characterization and pectin/nanochitosan nanocomposite characterization were presented as following:

The nanochitosan was synthesized by the ionic methacrylic acid method. The structure of the nanochitosan was studied by SEM at 100,000 and 150,000 magnifications and the acquired micrographs for nanochitosan film surface are shown in Figure 1. In this work, the particles are nearly spherical, the particle size diameters of the nanochitosan were less than 100 nm.

Figure 1. SEM micrograph of nanochitosan film.

FTIR analysis of NCS films

Figure 2 reports the spectra of nanochitosan films.  As can be known, the peaks recognized for the chitosan sample were related to C=O stretching amide I at 1635 cm-1 and to amide II at 1539 cm-1 [1]. These peaks slightly shifted to 1590 cm-1 and to 1550 cm-1 in the nanchitosan sample. The presence of intense band at 1590 cm-1 is attributed to the bond formed between the amino groups of chitosan and methacrylic acid.

3230

1590

1539

Figure 2. Infrared spectroscopy (FT-IR) spectra of nanochitosan film

[1] Woranuch S, Yoksan R., Eugenol-loaded chitosan nanoparticles: I. Thermal stability improvement of eugenol through encapsulation, Eugenol-loaded chitosan nanoparticles: I. Thermal stability improvement of eugenol through encapsulation, Carbohydr Polym. 2013 Jul 25;96(2):5, 96 (2013) 78-85.

Scanning electron microscopy (SEM)

      The SEM images of surface and cross-section of (P/NaCS means P:NaCS 50:50), NCS films are shown in Figure 3.

The surface of P:NCS 50:50 film is smothier than NCS film’s which indicates good compatibility between pectin and nanochitosan due to some interactions. The microstructure of this composition presents pores of small sizes across the surface of the film, thus showing the low water absorption, oxygen permeability and high tensile strength of P:NCS 50:50 film.  

Figure 3. SEM images of surface of P:NCS 50:50, NCS films (A, C respectively) and SEM images of cross-section of P:NCS 50:50, NCS films (B, D, respectively)

And this is a link related chitosan nanoparticle characterization and pectin/nanochitosan nanocomposite characterization (our previous research was published)

https://doi.org/10.3390/ijms21062224

Question 12: There is a lack of statistical test analysis in all measurements.

Answer: The statistical test analysis results were shown in “Data availability statement” file.

Question 13: Photography of the treated mangoes at different postharvest times would be interesting to include.

Answer: Photography of the treated mangoes at different postharvest times was not fully recorded. However, some special photos were recorded. Therefore, some photos of coated mangoes and uncoated mangoes were added in the manuscript. It reads now on page 10, line 341-345 and page 18, line 496-500. It was marked in red.

Conclusion

Question 14: Line 462...especially...grammar mistake

Answer: There is a mistake. It was changed to “moreover”. It reads now on page 22, line 581. It was marked in red.

Reviewer 2 Report

Biodegradable/food-grade polymers used as active coatings for shelf life extension of perishable fruits is a very relevant subject. The manuscript describes the use of coatings made from two biopolymers (pectin and chitosan) as strategies for extending the shelf life of mangoes (of a not very well-defined variety). The authors described a series of experiments that demonstrate that most of the physicochemical, sensory and microbiological characteristics of mango during storage are benefited from the use of the coating, in agreement with previous results. These results will be of interest mostly for mango supply chains, and academics dealing with active packaging and shelf life. The methodological approach was very simple and straightforward, and it did not involve robust instrumental analytical techniques, but, they allowed to gather enough evidence to demonstrate the effectiveness of the coating strategy. On the other hand, the tone of the manuscript is almost exclusively descriptive; the fact that no measurements of transport phenomena (oxygen permeation, ethylene production, etc.), nor of coating thicknesses, or coating gas barrier properties, left very little room for a more mechanistical discussion. I believe that a major drawback is that absolutely no statistical test was done to assess the significance of the differences observed regarding the effects of the different treatments. The graphical aspect and the excessive number of figures hinders the thorough, critical, detailed reading. Why there were no photographical evidence (which are so common and desirable in shelf life experiments! Almost customary). Some of the methodologies and the materials need more detailing. Also, a more focused comparison with the profuse literature regarding mango storage as affected by coatings, is recommendable.

To my concern, there are improvements that need to be done to the manuscript before publication.  Finally, the English language requires a thorough revision across the whole manuscript. My recommendation is to accept the work if authors do present a significantly improved version of it.

A detailed list of comments and corrections is presented below:

Title:

  • There might be a hyphen missing “Effect of Pectin/Nanochitosan-based Edible…”. Please revise.

Abstract:

  • The phrase “The mango fruits were im-30 mersed in different concentration 1.5%, 2.0%, 2.5% pectin/nanochitosan solution; 0.75%, 1% and 1.25% nanochitosan solution and stored at 17, 25 and 32°C for 24 days” is not easy to interpret, also because of the selected punctuation. Does it mean there were six treatments (3 with chitosan+pectin and 3 with only chitosan?) or that each one of the chitosan+pectin treatments varied the chitosan concentration i.e., that there were 9 treatments? Please rewrite to avoid the confusion.
  • Lines 32-33: “such as” is repeated, please rewrite.
  • Correct “viatmin C" in line 36.
  • “In the other words”; “storage temperature had influence on mango fruit quality and shelf life, so that the best temperature for storing of mango fruit was 17°C after 24 days and fruits stored at 25 and 39 32°C were destroyed after two weeks.” All this phrase is not scientifically sound, it is unclear, and it seems poorly written. Please revise and rewrite accordingly to express the findings cohesively and concisely.
  • The extension of shelf life of the best coating/temperature treatment tested must be clearly stated (in days) against the corresponding “blank” or control.

Introduction:

  • Line 48 “mangoes”… “they are harvested”.
  • Line 48 “…harvested, which limits their storage”
  • In general, the authors are highly advised to check the grammar and English style throughout all the manuscript.
  • Line 53: what does CA stand for? Define all acronyms.
  • To my concern, the definition of “nanochitosan” and the differences (structure, manufacturing, and morphology) of chitosan and nanochitosan must be better introduced in this section. This is particularly important since the coating was prepared from dissolved chitosan (and not “dispersed” chitosan nanoparticles). To my concern, authors should double check whether “nanoparticles solution” (in water?) is a correct way to describe such system. Usually, the expression “nanoparticle dispersion” would be more accurate.

Materials and methods:

  • Line 98: “nanochitosan particles were less than 100 nm”. Specify how this was ascertained at the experimental conditions.
  • Line 112: there must be separation between number and unit “2 g”, “98 ml”. Please check throughout the manuscript.
  • What chemicals were used to adjust pectin solutions to pH 4.5?
  • The pectin/nanochitosan coating solutions were prepared at 50:50 ratio of the two materials. This was not clear to me up to section 2.2. Please specify this in the abstract regarding this, as it is critical for future replications/verifications/comparisons.
  • More than 350 varieties of mango are harvested commercially in the world. Their fruits can be very different in terms of physiology and post-harvest behavior. Please provide more details regarding the agricultural variety of mango, or if possible, their genetic/cultivar information.
  • What was the average thickness of the coating after drying?
  • How many mangoes (replicas) were used for each treatment?
  • How did they mangoes were placed at each temperature? Any secondary packaging materials? What were the trays made of? What was the separation between samples during storage?

Results and discussion:

  • Line 177 is oddly written.
  • The acronyms used in the figures must be defined in the corresponding caption; I am assuming P/NaCS means pectin plus nanochitosan, nbut it should be stated. Also the meaning of error bars (n=?). The graphical aspect of the Figure 1 is not very appealing, perhaps consider using more uniform/professional lines, and invisible external linings.
  • Of all the measurements described in section 2.4, which are the ones showed in Figure 2? (which probe? Which position?)…Same for Figure 6.
  • Lines 193-194: “the reason might be due to the coating and coating thickness which affected the transpiration and respiration rates of the mango”. But neither thickness nor transpiration rates were measured, apparently.
  • Lines 229-232: All this information is not relevant whatsoever in the section of results and discussion. If much, they should be transferred to the corresponding methodological section.
  • Figure 3 is very difficult to review. The space proportions within the charts, the letter indications, the separation, and the fact that is extends itself over three pages, are not helpful to readers.
  • Line 246: “L* value of mango flesh of the sample coated with 2% pectin/nanochitosan increased slowly during 15 days of storage. The 2.5% pectin/nanochitosan coatings were negative effect on the mango color, namely the L* value decreased significantly at the end of the storage time.” The comments on this observation are not explicative of this difference. Why does such difference
  • The discussion of the results of instrumental colorimetry, as presented, is quite confusing and unnecessarily intricate and long; these data require a different statistical approach, a multivariate one, and a more concise discussion. If authors are not able to perform a PCA or a similar technique, I believe it would be much more informative to provide a DeltaE plot, and basing the further critical analysis upon this one information. Perhaps L coordinate and DeltaE.
  • A major drawback is that absolutely no statistical test was done to assess the significance of the differences.
  • “significantly effective” in this context seems loose. Line 333. “was not significantly different from”. Line 225. Etc.
  • How these results compare to previous studies done using chitosan coatings in mangoes?:
  • Figure 8 is also very difficult to read; it extends for 3 pages unnecessarily. Please correct this because it can be somewhat distracting. It is, as almost all the figures, actually at least three figures, which makes for an excessive number of figures, more appropriate for other (loner) types of documents, as dissertations. The figure corresponding to %TSS seems to be repeated.

https://doi.org/10.1111/j.1745-4549.2008.00213.x

https://en.cnki.com.cn/Article_en/CJFDTotal-SSPJ200703069.htm

https://en.cnki.com.cn/Article_en/CJFDTotal-SSPJ200703069.htm

https://doi.org/10.1016/j.lwt.2020.110809

and so many others

  • Along with the excessive number of figures, a photographical evidence is very much missing.

Author Response

The authors thank the editor and the reviewers for reviewing this manuscript and for your comments and suggestions, which greatly improved the quality of the paper. Detailed responses to the comments are provided below.

Reviewer 2 Comments

The authors thank the reviewer for taking the time to review the manuscript. Detailed responses to the comments are provided below.

English language and style

( ) Extensive editing of English language and style required
(x) Moderate English changes required
( ) English language and style are fine/minor spell check required
( ) I don't feel qualified to judge about the English language and style

Yes

Can be improved

Must be improved

Not applicable

Does the introduction provide sufficient background and include all relevant references?

( )

(x)

( )

( )

Is the research design appropriate?

( )

(x)

( )

( )

Are the methods adequately described?

( )

(x)

( )

( )

Are the results clearly presented?

( )

( )

(x)

( )

Are the conclusions supported by the results?

( )

(x)

( )

( )

Comments and Suggestions for Authors

Question 1: Authors are writing about "chitosan-polymethacrylic nanoparticles" - is PMMA actually edible? Because I do not think so, since it is an engineering plastic.

Answer: Thank you for your suggestion. When we did this research, we thought we could use PMMA for fruits coatings because methacrylic acid and K2S2O8 were used with a low amount. Chitosan nanoparticles were twice dissolved in distilled water. In the future, we’ll do the research for nanochitosan production using safer and cleaner method. In this study, we already omitted “edible” from the title.

This is a link related Potassium persulfate: Food additives permitted for direct addition to food for human - Coatings on fresh citrus fruit.

 https://www.accessdata.fda.gov/scripts/cdrh/cfdocs/cfCFR/CFRSearch.cfm?fr=172.210

Component

Limitations

Potassium persulfate

Do.

And this is a link related anionic methacrylate copolymer for the proposed uses as a food additive

https://efsa.onlinelibrary.wiley.com/doi/epdf/10.2903/j.efsa.2010.1656

Question 2: "The reason for the difference in weight loss among samples might be due to the coating and coating thickness which affected the transpiration and respiration rates of the mango" - what was the coating thickness for particular variants? Same question for oxygen permeability since Authors are writing about respiration rates.

Answer: Before doing this research, we referred the paper “Measurement and modeling the effect of temperature, relative humidity and storage duration on the transpiration rate of three banana cultivars”, we calculated and predicted transpirate of mango at 25 oC and 75% relative humidity the corresponding mango surface is required for choosing the coating. The information is presented as following table.

Table 1. Water vapor permeability (WVP) and Water vapor transmission rate (WVTR) of coatings

Type of coating

Water vapor permeability (WVP), g.mm/m2.day.kPa

Water vapor transmission rate (WVTR), g/m2.day

2% P/NaCS coating (~15,0±2,0 µm)

0.653±0.039

25.77±1.05

Table 2.  Oxygen permeability (OP) and oxygen transmission rate (OTR) of coatings

Type of coating

Oxygen permeability (OP), cc.mm/m2.day

Oxygen transmission rate (OTR),

cc/m2.day

2% P/NaCS coatings (~15,0±2,0 µm)

151.6±14.4

59.2±9.9

Table 3. Weight loss,  Required water vapour transmission rate, Required oxygen

of mango fruit at 25 oC

Type of fruits

Weight of a mango fruit, kg

Surface square of a mango fruit, cm2

Weight loss, g

Required water vapour transmission rate, g/m2.day

Required oxygen, cc/m2.day

Mango fruits

0.361±0.004

242.2±7,7

23,71±0.46

335.63±3.61

4500.31±110.21

These results indicated that coatings can be used as water vapor barrier and oxygen barrier for mango fruits preservation.

Reference: Murmu, S.B., & Mishra, H. N., Measurement and modeling the effect of temperature, relative humidity and storage duration on the transpiration rate of three banana cultivars. Scientia Horticulturae 2016. 209: p. 124-131.

Question 3: ". The coating maintaining the fruit firmness could probably be explained by the coating inhibiting moisture loss and delaying the degradation of insoluble protopectins to soluble pectin and pectic acid [28], and other reactions induced by oxygen" - same as above, please present the oxygen and moisture permeability of coatings.

Answer: In our previous research, oxygen permeability (OP) and water vapor permeability (WVP) of coatings were determined as following tables. This information was added in the manuscript to explain the results. It reads now on page 5, lines 233-237. It was marked in red. 

Table 4. Water vapor permeability and Water vapor transmission rate of films

Films

Water vapor permeability,

g.mm/m2.day.kPa

Water vapor transmission rate,

 g/m2.d

P:NaCS (50:50)

0.2052±0.0083d

8.10±0.33d

NaCS

0.1755±0.0085d

9.24±0.45cd

Table 5. Oxygen permeability (OP) and oxygen transmission rate (OTR)

Films

Oxygen permeability,

cc.mm/m2.day

oxygen transmission rate,

cc/m2.day

P:NaCS (50:50)

47.67±5.11d

18.63±2.17d

NaCS

832.23±49.89c

320.8±25.88c

Question 4: For the color changes, except for graphs please present the actual colors, since the journal publishes in color. It will be significantly better and easier to see the actual changes.

Answer: Thank you for your suggestion. The photographs of mangoes were added in the manuscript to present the actual colors. It reads now on page 10, line 341-345 and page 18, line 496-500. It was marked in red.

Question 5: In Figure 3 there is no parts g and h presented, ends with f.

Answer: I am sorry for the inconvenience. Figure 3 extends 3 pages so it is difficult to read. “a”, “b”, “c”, “d”, “e”, “f”, “g” and “h” letters were presented below each chart.

Question 6: Why 2% P/NaCS sample was selected for further analysis, not the 2.5 since it showed better results?

Answer: Thank you for your suggestion. The reason for choosing 2% P/NaCS sample was rewritten. It reads now on page 12-13, line 376-390. It was marked in red.

Question 7: Please refer more to the literature reports during description of results. (Need more discussion and references)

Answer: Thank you for your suggestion. Some discussion and references were added in the Results and Discussion Part. It was marked in red.

Reviewer 3 Report

Authors are writing about "chitosan-polymethacrylic nanoparticles" - is PMMA actually edible? Because I do not think so, since it is an engineering plastic.

"The reason for the difference in weight loss among samples might be due to the coating and coating thickness which affected the transpiration and respiration rates of the mango" - what was the coating thickness for particular variants? Same question for oxygen permeability since Authors are writing about respiration rates.

". The coating maintaining the fruit firmness could probably be explained by the coating inhibiting moisture loss and delaying the degradation of insoluble protopectins to soluble pectin and pectic acid [28], and other reactions induced by oxygen" - same as above, please present the oxygen and moisture permeability of coatings.

For the color changes, except for graphs please present the actual colors, since the jounral publishes in color. It will be significantly better and easier to see the actual changes.

In Figure 3 there is no parts g and h presented, ends with f.

Why 2% P/NaCS sample was selected for further analysis, not the 2.5 since it showed better results?

Please refer more to the literature reports during description of results.

Author Response

The authors thank the editor and the reviewers for reviewing this manuscript and for your comments and suggestions, which greatly improved the quality of the paper. Detailed responses to the comments are provided below.

Reviewer 3 Comments

The authors thank the reviewer for taking the time to review the manuscript. Detailed responses to the comments are provided below.

English language and style

( ) Extensive editing of English language and style required
(x) Moderate English changes required
( ) English language and style are fine/minor spell check required
( ) I don't feel qualified to judge about the English language and style

Yes

Can be improved

Must be improved

Not applicable

Does the introduction provide sufficient background and include all relevant references?

( )

(x)

( )

( )

Is the research design appropriate?

(x)

( )

( )

( )

Are the methods adequately described?

( )

( )

(x)

( )

Are the results clearly presented?

( )

( )

(x)

( )

Are the conclusions supported by the results?

(x)

( )

( )

( )

Comments and Suggestions for Authors

Biodegradable/food-grade polymers used as active coatings for shelf life extension of perishable fruits is a very relevant subject. The manuscript describes the use of coatings made from two biopolymers (pectin and chitosan) as strategies for extending the shelf life of mangoes (of a not very well-defined variety). The authors described a series of experiments that demonstrate that most of the physicochemical, sensory and microbiological characteristics of mango during storage are benefited from the use of the coating, in agreement with previous results. These results will be of interest mostly for mango supply chains, and academics dealing with active packaging and shelf life. The methodological approach was very simple and straightforward, and it did not involve robust instrumental analytical techniques, but, they allowed to gather enough evidence to demonstrate the effectiveness of the coating strategy. On the other hand, the tone of the manuscript is almost exclusively descriptive; the fact that no measurements of transport phenomena (oxygen permeation, ethylene production, etc.), nor of coating thicknesses, or coating gas barrier properties, left very little room for a more mechanistical discussion. I believe that a major drawback is that absolutely no statistical test was done to assess the significance of the differences observed regarding the effects of the different treatments. The graphical aspect and the excessive number of figures hinders the thorough, critical, detailed reading. Why there were no photographical evidence (which are so common and desirable in shelf life experiments! Almost customary). Some of the methodologies and the materials need more detailing. Also, a more focused comparison with the profuse literature regarding mango storage as affected by coatings, is recommendable.

To my concern, there are improvements that need to be done to the manuscript before publication.  Finally, the English language requires a thorough revision across the whole manuscript. My recommendation is to accept the work if authors do present a significantly improved version of it.

A detailed list of comments and corrections is presented below:

Question 1:

Title: There might be a hyphen missing “Effect of Pectin/Nanochitosan-based Edible…”. Please revise.

 Answer: Thank you for your recommendation. The hyphen is added. The title is corrected into “Effect of Pectin/Nanochitosan-based Edible Coatings and Storage Temperature on Shelf-life Extension of “Elephant” Mango (Mangifera Indica L.) Fruit”. It reads now on page 1, line 2. It was marked in red.

Abstract:

Question 2:

The phrase “The mango fruits were im-30 mersed in different concentration 1.5%, 2.0%, 2.5% pectin/nanochitosan solution; 0.75%, 1% and 1.25% nanochitosan solution and stored at 17, 25 and 32°C for 24 days” is not easy to interpret, also because of the selected punctuation. Does it mean there were six treatments (3 with chitosan+pectin and 3 with only chitosan?) or that each one of the chitosan+pectin treatments varied the chitosan concentration i.e., that there were 9 treatments? Please rewrite to avoid the confusion.

Answer: Thank you for your suggestion. The experimental design is rewritten. It reads now on page 3-4, lines 133-150. It was marked in red. It means there were six treatments (3 with nanochitosan+pectin and 3 with only nanochitosan)

Question 3: Lines 32-33: “such as” is repeated, please rewrite.

Answer: ‘’including’’ is instead of ‘’such as’’. It reads now on page 1, line 35. It was marked in red.

Question 4:

Correct “viatmin C" in line 36.

Answer: “Vitamin C” is corrected. It reads now on page 1, line 39. It was marked in red.

Question 5:

“In the other words”; “storage temperature had influence on mango fruit quality and shelf life, so that the best temperature for storing of mango fruit was 17°C after 24 days and fruits stored at 25 and 39 32°C were destroyed after two weeks.” All this phrase is not scientifically sound, it is unclear, and it seems poorly written. Please revise and rewrite accordingly to express the findings cohesively and concisely.

Answer: This sentence is rewritten. It reads now on page 1, lines 40-42. It was marked in red.

Question 6:

The extension of shelf life of the best coating/temperature treatment tested must be clearly stated (in days) against the corresponding “blank” or control.

 Answer: Thank you for your suggestion. This result is added in the abstract. It reads now on page 1, lines 40-42. It was marked in red.

Introduction:

Question 7:

Line 48 “mangoes”… “they are harvested”.

Answer:  There is a mistake. It was changed into “they are”. It reads now on page 2, line 52. It was marked in red.

Question 7:

Line 48 “…harvested, which limits their storage”

Answer: There is a mistake. It was changed into “which limit their storage”. It reads now on page 2, line 52. It was marked in red.

In general, the authors are highly advised to check the grammar and English style throughout all the manuscript.

Answer: Thank you for your suggestion. We already tried to check the grammar and English style throughout all the manuscript and corrected them.

Question 8:

Line 53: what does CA stand for? Define all acronyms.

Answer: There is a mistake. It was changed. It reads now on page 2, line 58-59. It was marked in red.

Question 9:

To my concern, the definition of “nanochitosan” and the differences (structure, manufacturing, and morphology) of chitosan and nanochitosan must be better introduced in this section. This is particularly important since the coating was prepared from dissolved chitosan (and not “dispersed” chitosan nanoparticles). To my concern, authors should double check whether “nanoparticles solution” (in water?) is a correct way to describe such system. Usually, the expression “nanoparticle dispersion” would be more accurate.

Answer: Thank you for your suggestion. “Nanochitosan solution” was changed into “nanochitosan dispersion” on the whole manuscript. Some information about chitosan nanoparticle characterization was added in the manuscript (2.1 Materials section). It reads now on page 3, lines 105-108. They were marked in red.

Materials and methods:

Question 10: Line 98: “nanochitosan particles were less than 100 nm”. Specify how this was ascertained at the experimental conditions.

Answer: The reference is added in this sentence. Some information about chitosan nanoparticle characterization was added in the manuscript (2.1 Materials section). It reads now on page 3, lines 105-108. It was marked in red.

In our previous research, the SEM of nanochitosan was taken as following. The FTIR of nanochitosan was also presented.

Figure 1. SEM micrograph of nanochitosan film.

Figure 2 reports the spectra of nanochitosan films.  As can be known, the peaks recognized for the chitosan sample were related to C=O stretching amide I at 1635 cm-1 and to amide II at 1539 cm-1 [1]. These peaks slightly shifted to 1590 cm-1 and to 1550 cm-1 in the nanchitosan sample. The presence of intense band at 1590 cm-1 is attributed to the bond formed between the amino groups of chitosan and methacrylic acid.

3230

1590

1539

Figure 2. Infrared spectroscopy (FT-IR) spectra of nanochitosan film

Question 11: Line 112: there must be separation between number and unit “2 g”, “98 ml”. Please check throughout the manuscript.

Answer: Thank you for your suggestion. I already checked and corrected them throughout the manuscript. They were marked in red.

Question 12: What chemicals were used to adjust pectin solutions to pH 4.5?

Answer: 2 M Na2CO3 was used to adjust pectin solutions to pH 4.5. I also added this chemical in the manuscript. It reads now on page 3, line 129. It was marked in red.

Question 13: The pectin/nanochitosan coating solutions were prepared at 50:50 ratio of the two materials. This was not clear to me up to section 2.2. Please specify this in the abstract regarding this, as it is critical for future replications/verifications/comparisons.

Answer: Thank you for your suggestion. The ratio of pectin:nanochitosan 50:50 is added in the abstract. It reads now on page 1, lines 33-34. It was marked in red.

Question 14: More than 350 varieties of mango are harvested commercially in the world. Their fruits can be very different in terms of physiology and post-harvest behavior. Please provide more details regarding the agricultural variety of mango, or if possible, their genetic/cultivar information.

Answer: “Elephant” mango variety of mango was used for this research. The image of “Elephant” mango was added in the manuscript. It reads now on page 3, line 114.

Question 15: What was the average thickness of the coating after drying?

Answer: The average thickness of the coating after drying was about 15 μm with Concentration of pectin/nanochitosan dispersion: 2% w/v. This information was added in 2.2.1 section. It reads now on page 3, lines 138-139. It was marked in red.

Question 16: How many mangoes (replicas) were used for each treatment?

How did they mangoes were placed at each temperature? Any secondary packaging materials? What were the trays made of? What was the separation between samples during storage?

Answer: Thank you for your suggestion. Three mangoes were sampled for analysis each time for each treatment. Mangoes were placed onto cardboard trays by hand. All the fruits were stored at different temperature with relative humidity of 75%. The experimental treatments were rewritten. It reads now on page 3, lines 133-150. It was marked in red.

 Results and discussion:

Question 17: Line 177 is oddly written.

Answer: There is a mistake. It was changed. It reads now on page 5, lines 198-199. It was marked in red.

Question 18: The acronyms used in the figures must be defined in the corresponding caption; I am assuming P/NaCS means pectin plus nanochitosan, nbut it should be stated. Also the meaning of error bars (n=?). The graphical aspect of the Figure 1 is not very appealing, perhaps consider using more uniform/professional lines, and invisible external linings.

Answer: Thank you for your suggestion. We already corrected Figure 1 and other figures according to your comments. We already explained the meaning of error bars (n=3) all the figures.

Question 19: Of all the measurements described in section 2.4, which are the ones showed in Figure 2? (which probe? Which position?)…Same for Figure 6.

Answer: This description is added. “Fruit firmness was measured using a TA. XT plus texture analyzer with a 2 mm diameter and 25 mm length puncture probe, calibrated with a 5 kg load cell. Initial grip separation was set at 30 mm with a test speed of 5 mm/s and a depth of 5 mm. The maximum force (N) was measured at 3 positions - basal, middle, and upper positions of each fruit. Fruit firmness value of each fruit was calculated as mean of 3 measurements”.

It reads now on page 4, lines 158-162. It was marked in red.

Below is equipment for testing firmness.

Question 20: Lines 193-194: “the reason might be due to the coating and coating thickness which affected the transpiration and respiration rates of the mango”. But neither thickness nor transpiration rates were measured, apparently.

Answer: Some information about thickness and oxygen permeability of coating was added in the manuscript. It reads now on page 5, lines 233-237. It was marked in red.

Question 21: Lines 229-232: All this information is not relevant whatsoever in the section of results and discussion. If much, they should be transferred to the corresponding methodological section.

Answer: Thank you for your suggestion. All this information is transferred to the corresponding methodological section.  It reads now on page 4, lines 170-174. It was marked in red.

Question 22: Figure 3 is very difficult to review. The space proportions within the charts, the letter indications, the separation, and the fact that is extends itself over three pages, are not helpful to readers.

The discussion of the results of instrumental colorimetry, as presented, is quite confusing and unnecessarily intricate and long; these data require a different statistical approach, a multivariate one, and a more concise discussion. If authors are not able to perform a PCA or a similar technique, I believe it would be much more informative to provide a DeltaE plot, and basing the further critical analysis upon this one information. Perhaps L coordinate and DeltaE.

Answer: Thank you for your suggestion. It is quite difficult for us to write the whole section. We already tried to correct some sentences and added the images in this section according to reviewer’s comments. We will make a further effort to write color analysis according to your comments and suggestions in the future. Some photographs of mangoes were added in the manuscript to present the actual colors. It reads now on page 10, line 341-345 and page 18, line 496-500.

Question 23: Line 246: “L* value of mango flesh of the sample coated with 2% pectin/nanochitosan increased slowly during 15 days of storage. The 2.5% pectin/nanochitosan coatings were negative effect on the mango color, namely the L* value decreased significantly at the end of the storage time.” The comments on this observation are not explicative of this difference. Why does such difference

Answer: We already added the actual photos of mangoes to clear the change of color. It reads now on page 10, line 341-345 and page 18, line 496-500. The reason of difference is also explained in the line 376-390, on page 12, 13.

Question 24: A major drawback is that absolutely no statistical test was done to assess the significance of the differences.

Answer: All results were analyzed statistically and we already had standard deviation in the graph. It is difficult for us to show the significant differences between group samples so that we send the Supplemental Data in the attach file “Data availability statement”.

Question 25: “Data availability statement” file was added..

Answer: These phrases were rewritten to present the meaning clearly. They were marked in red.

Question 26: How these results compare to previous studies done using chitosan coatings in mangoes?

https://doi.org/10.1111/j.1745-4549.2008.00213.x

https://en.cnki.com.cn/Article_en/CJFDTotal-SSPJ200703069.htm

https://en.cnki.com.cn/Article_en/CJFDTotal-SSPJ200703069.htm

https://doi.org/10.1016/j.lwt.2020.110809

and so many others

Answer: Thank you for your recommendation. I already read these references related chitosan coating to rewrite the comparison between chitosan coating and pectin/nanochitosan coating. It reads now on page 21, lines 562-568. It was marked in red.

Question 27: Figure 8 is also very difficult to read; it extends for 3 pages unnecessarily. Please correct this because it can be somewhat distracting. It is, as almost all the figures, actually at least three figures, which makes for an excessive number of figures, more appropriate for other (loner) types of documents, as dissertations. The figure corresponding to %TSS seems to be repeated.

Answer: Thank you for your suggestion. Reducing sugar is omitted from Figure 8. All the figures are changed to understand more easily.

Question 28: Along with the excessive number of figures, a photographical evidence is very much missing.

Answer: The photographs of mangoes were added in the manuscript to present the actual colors. It reads now on page 10, line 341-345 and page 18, line 496-500.

Round 2

Reviewer 1 Report

Dear authors,

The word nanochitosan must be corrected across the whole work. It is not well-written.

The statistical analysis is still missing. You will need to use some statistical test to compare means, e.g. Duncan, Tukey, Kruskal-Wallis, etc. It is mandatory to do this in present work. 

Author Response

The authors thank the editor and the reviewers for reviewing this manuscript and for your comments and suggestions, which greatly improved the quality of the paper.

Authors thank the editor for your kind supporting. Authors are sorry for the inconvenience.

Detailed responses to the comments are provided below.

Reviewer 1 Comments

The authors thank the reviewer for taking the time to review the manuscript. Detailed responses to the comments are provided below.

Open Review

English language and style

( ) Extensive editing of English language and style required
( ) Moderate English changes required
( ) English language and style are fine/minor spell check required
(x) I don't feel qualified to judge about the English language and style

Yes

Can be improved

Must be improved

Not applicable

Does the introduction provide sufficient background and include all relevant references?

(x)

( )

( )

( )

Is the research design appropriate?

( )

(x)

( )

( )

Are the methods adequately described?

( )

(x)

( )

( )

Are the results clearly presented?

( )

(x)

( )

( )

Are the conclusions supported by the results?

(x)

( )

( )

( )

Comments and Suggestions for Authors

Question: The word nanochitosan must be corrected across the whole work. It is not well-written.

Answer: Thank you for your suggestion. The words “nanochitosan” were corrected across the whole work. According to your question and our understanding, authors corrected some words such as “chitosan nanoparticle”, “chitosan nanoparticles” into “nanochitosan” across the whole manuscript. It reads now on page 3, line 105-108 and page 6, line 243. It was marked in red.

Question: The statistical analysis is still missing. You will need to use some statistical test to compare means, e.g. Duncan, Tukey, Kruskal-Wallis, etc. It is mandatory to do this in present work. 

Answer: Thank you for your suggestion. The statistical analysis was added in the manuscript “All data were analyzed by one-way ANOVA. Mean separation was performed by Duncan’s multiple range tests with significance level (p≤0.05). Statistical analysis was carried out by using Minitab16 software (Version 8.0, Northampton, USA)”. It reads now on page 5, line 210-212. It was marked in red.

Reviewer 3 Report

Unfortunately, I have received the Answers to other Reviewer comments in the file attached by Authors so I am not able to make the decision. 

Author Response

(The authors gave the same response as above.)

Round 3

Reviewer 3 Report

Once again. Unfortunately, I have received the Answers to other Reviewer comments in the file attached by Authors so I am not able to make the decision. 

My comments from first review, which were not adressed yet are presented below:

Authors are writing about "chitosan-polymethacrylic nanoparticles" - is PMMA actually edible? Because I do not think so, since it is an engineering plastic.

"The reason for the difference in weight loss among samples might be due to the coating and coating thickness which affected the transpiration and respiration rates of the mango" - what was the coating thickness for particular variants? Same question for oxygen permeability since Authors are writing about respiration rates.

". The coating maintaining the fruit firmness could probably be explained by the coating inhibiting moisture loss and delaying the degradation of insoluble protopectins to soluble pectin and pectic acid [28], and other reactions induced by oxygen" - same as above, please present the oxygen and moisture permeability of coatings.

For the color changes, except for graphs please present the actual colors, since the jounral publishes in color. It will be significantly better and easier to see the actual changes.

In Figure 3 there is no parts g and h presented, ends with f.

Why 2% P/NaCS sample was selected for further analysis, not the 2.5 since it showed better results?

Please refer more to the literature reports during description of results.

Round 4

Reviewer 3 Report

"

Question 4: For the color changes, except for graphs please present the actual colors, since the journal publishes in color. It will be significantly better and easier to see the actual changes.

Answer: Thank you for your suggestion. The photographs of mangoes were added in the manuscript to present the actual colors in Figure 5 and Figure 10."

I would recommend to include the Table with the digital reproduction of color since Authors have all color parameters. It would be very helpful, even as supplementary material.